# Diagnostic Excellence in Pediatric Spine Imaging: Using Contextualized Imaging Protocols

**DOI:** 10.3390/diagnostics13182973

**Published:** 2023-09-18

**Authors:** Nadja Kadom, Kartik Reddy, Maxwell E. Cooper, Jack Knight-Scott, Richard A. Jones, Susan Palasis

**Affiliations:** 1Department of Radiology and Imaging Sciences, Emory University School of Medicine, Atlanta, GA 30322, USA; 2Department of Radiology, Children’s Healthcare of Atlanta, Atlanta, GA 30342, USA

**Keywords:** radiology, pediatric, imaging, spine, mri protocol, protocoling, contextual

## Abstract

Contextual design and selection of MRI protocols is critical for making an accurate diagnosis given the wide variety of clinical indications for spine imaging in children. Here, we describe our pediatric spine imaging protocols in detail, tailored to specific clinical questions.

## 1. Introduction

Delivering the right treatment to the right patient at the right time is a key principle in Personalized Medicine [1]. The radiology equivalent to this concept would be delivering the right imaging study to the right patient at the right time. Performing the wrong procedure can lead to missed, delayed, or wrong diagnoses, resulting in potential patient harm and/or medicolegal actions against radiologists [2]. Diagnostic error represents a considerable issue in the United States. Approximately 12 million U.S. adults experience outpatient diagnostic errors annually [3]. In 48% of malpractice claims against radiologists there was a misinterpretation that resulted in a delayed diagnosis and/or treatment of the patient [4].

It is conceivable that performing an imaging test on a child with an imaging protocol that is not tailored to the clinical question can result in a missed or inaccurate diagnosis. Children are affected by different diseases compared to adults, and their anatomy and physiology differ while undergoing development [5]. For example, the main indications for magnetic resonance imaging (MRI) of the spine in adults is evaluation of back pain, neoplastic disease, trauma, and cord compression; in children, the indications vary much more widely and include scoliosis, congenital malformations, gait abnormalities, and Chiari I malformation [6,7]. Therefore, it may be worthwhile expanding the range of imaging protocols for practices that serve pediatric patients.

The term “contextualized” was first used in radiology reporting templates for a specific disease or indication [8]. The term may be similarly applied to MRI protocols that are tailored to specific imaging indications. For example, at our institution, a lumbar spine MRI protocol in a patient with suspected cord tethering is different from a lumbar spine MRI protocol for a patient with lumbar pain or a lumbar vertebral body tumor.

We developed a set of contextualized spine imaging protocols for use in pediatric populations. Here, we assessed the breadth of spine imaging indications at our institution and shared a detailed description and rationale for our MRI protocols.

## 2. Methods

The study was IRB-exempt and HIPAA-compliant. Our institution is a freestanding pediatric care system with two hospitals that serve as level 1 trauma centers and several outpatient imaging centers. We searched the radiology information system (RIS) for all MRI spine imaging studies from 1 July 2021 to 30 June 2022 using a commercial search tool (mPower^TM^, Nuance, Burlington, MA, USA).

Duplicate patient names were removed from the data set to improve the accuracy of represented clinical indications. This prevented us, however, from identifying combination studies, such as cervical/thoracic or thoracic/lumbar MRI.

Indications were coded by a board-certified pediatric neuroradiologist (N.K.) based on clinical categories. Indications of “Chiari” could not be differentiated as Chiari I versus Chiari II. Each ordering provider unknown to the coder was looked up using an Internet search engine to determine their medical specialty. It was difficult to code a variety of non-specific clinical symptoms in the exam indications that could occur with many underlying pathologies. As an example, clinical signs of sacral dimple, a palpable sacral mass, or a gluteal cleft asymmetry, urinary dysfunction, bowel dysfunction, back pain, lower extremity weakness, and scoliosis can all be associated with cord tethering [9]. In those cases, the coder used their professional judgment, considering the patient age, the provider specialty, and the imaged anatomy. For example, lumbar spine MRIs ordered by urology for urinary dysfunction were coded as “Tethering”, while spine MRIs ordered by neurology for weakness were coded as “Weakness” and those for scoliosis ordered by an orthopedic surgeon were coded as “Scoliosis”.

We used descriptive statistics (Stata^(R)^ v13.1 StataCorp LP, College Station, TX, USA) to report frequencies of study indications, imaging protocols, and basic demographic information, such as patient age, gender, and type (emergency, inpatient, outpatient).

We provide a detailed description and rationale for each of our contextualized pediatric spine imaging protocols with illustrative examples.

## 3. Results

Demographics. During the study period a total of 12,741 MRI studies were performed for neuroimaging indications and 2324 (18%) were MRI studies of the spine. We imaged the spine in 1221 (53%) female and 1103 (47%) male patients. The mean patient age was 9.9 years (range: 1 day to 24 years). The majority were outpatient studies (*n* = 1794, 77%), followed by inpatient studies (*n* = 490, 21%) and emergency studies (*n* = 40, 2%).

Anatomy. The following parts of the spine were imaged: complete spine (*n* = 1296, 56%), cervical spine (*n* = 486, 21%), thoracic spine (*n* = 0), lumbar spine (*n* = 542, 23%). Of note, in a similar analysis we performed in 2019 the incidence of isolated thoracic spine imaging was 0.3% (*n* = 7). Studies were performed without IV contrast (*n* = 1431, 62%) and with IV contrast (*n* = 893, 38%). Among studies with IV contrast, 516 (22%) were performed following a non-contrast MRI, while 377 (16%) were performed solely post-contrast.

Indications. There were 28 different indications based on the coding process (Figure 1). Among the top ten indications were CNS tumor (*n* = 516, 22%), pain (*n* = 351, 15%), cord tethering (*n* = 277, 12%), abnormal spinal curve (*n* = 189, 8%), Chiari (*n* = 135, 6%), inflammation (*n* = 125, 5%), trauma (*n* = 111, 5%), weakness (*n* = 93, 4%), non-accidental trauma (*n* = 68, 3%), and abnormal gait (*n* = 57, 2%).

Ordering Providers. There were 20 specialties among the ordering providers (Figure 1). Most imaging orders came from neurosurgery (*n* = 522, 23%) and orthopedics (*n* = 522, 23%), followed by hematology/oncology (*n* = 416, 17%), pediatrics (*n* = 326, 14%), and neurology (*n* = 298, 13%).

Among the top ten indications for pediatric spine imaging in our cohort were seven indications for which we have a contextualized MRI spine protocol: CNS tumor, pain, cord tethering, abnormal curve, Chiari I, trauma, and non-accidental trauma (Figure 2).

For all our spine MR imaging studies of the thoracic and/or lumbar spine we include a sagittal image to count all vertebral levels from top to bottom. This practice acknowledges the possibility for transitional anatomy and our desire to avoid wrong-site surgery [10]. All our pediatric spine MRI protocols align with best practice parameters and are continually updated [11].

CNS tumor. Most spinal imaging in our cohort was to screen for spinal metastatic disease in patients with brain tumors. The incidence of primary spinal cord tumors is only 10% of all pediatric CNS tumors [12]. Our spine MRI protocol for patients with CNS tumors follows evidence-based recommendations from experts in the field of pediatric oncologic imaging and are continually updated [12,13]. Our contextualized modification of the CNS tumor imaging protocol entails a first-time CNS tumor imaging protocol with pre- and post-contrast imaging of the brain and post-contrast imaging of the spine, and continued omission of any pre-contrast spine imaging for follow-up exams. This serves to shorten exam times and can decrease the need for/duration of sedation in the pediatric population [14,15].

Imaging signs of spinal leptomeningeal disease include abnormal leptomeningeal enhancement and/or extramedullary nodules, which may or may not be enhancing. Our MRI protocol includes two 3D sequences, one post-contrast 3D T1-weighted sequence to assess for enhancing lesions [16], and one heavily T2-weighted to detect small nodular leptomeningeal disease that may not be enhancing [17] (Table 1) (Figure 3). Primary brain tumors that are diffusion-restricting can benefit from the addition of spine diffusion to detect non-enhancing cellular metastatic lesions on spine surveillance MR imaging [13].

Pain. The prevalence of back pain in children varies by age and country studied but can be as high as 51% [18]. Imaging is not indicated in children with a short duration of pain, normal physical examination, and no history of trauma [19]. Imaging is utilized in children with back pain when there is either suspicion for serious pathologies, such as fracture, tumor, or infection, or when there is chronic pain that is not responsive to medical management [19].

Spondylolysis is a common cause of back pain in children and adolescents, found in 12–16% [18]. The diagnosis of spondylolysis is made by radiography and computed tomography (CT) imaging [18]. However, early signs of spondylolysis, such as pedicle edema, are not apparent on CT but can be detected with MR imaging. Detection of pedicle edema from spondylolysis can help establish a diagnosis early and avoid the need for radiographs and CT imaging, which require exposing patients to ionizing radiation [20]. Our contextualized MRI protocol for lumbar back pain includes standard sequences for the evaluation of intervertebral discs and neural foramina and a sagittal T2-weighted sequence with fat suppression (STIR) to detect pedicle edema, as well as a 3D T1-weighted sequence for assessing integrity of the pars articularis (Table 2) (Figure 4).

Cord tethering. The incidence of tethered cord syndrome is estimated to be 12 per 100,000 children [9]. Tethered cord syndrome (TCS) is a condition where there is tension on the spinal cord, either due to the cord tethering to an intraspinal lesion or due to an abnormally tight terminal filum [9]. MRI is best suited to detect signs of cord tethering and identify lesions that could be managed surgically to avoid the long-term effects of ischemic injury, as well as motor and sensory deficits [9].

Many instances of cord tethering demonstrate abnormal fatty tissue within the filum terminale [9]. Cord tethering may be present when the tip of the conus is located below the L2/3 disc level. Correct determination of the level of the conus requires counting spinal levels from top to bottom, to account for possible transitional anatomy. Our contextualized lumbar MRI protocol for cord tethering includes a 3D T1-weighted sequence [21] for the detection of fatty tissue within the spinal canal and terminal filum (Table 3) (Figure 5).

Abnormal spinal curve. Scoliosis affects ~3% percent of the general population and can develop in infancy or early childhood [22]. Scoliosis is idiopathic in 60% of patients and may have an underlying cause in the remaining 40%, such as cerebral palsy, myelomeningocele, tethered cord syndrome, spinal muscular atrophy, syringomyelia, muscular dystrophy, musculoskeletal disorders, connective tissue disorders, intra- or paraspinal mass lesions, or genetic syndromes [23].

Radiography is the recommended initial imaging modality for the diagnosis and surveillance of scoliosis [19]. CT can be obtained to provide better visualization of complex osseous deformities and aid in surgical planning [19]. MRI is indicated in patients with concerning clinical presentations and serves to evaluate for intraspinal abnormalities, such as tumors, syringomyelia, spinal dysraphism, cord tethering, or bony pathology, or extraspinal causes of scoliosis, such as paraspinal tumors [19].

The typical MRI sequences for scoliosis patients include sagittal and axial T1- and T2-weighted imaging [19]. Fat suppression has several uses for scoliosis patients, including confirming congenital fatty lesions [19]. In addition, fat-suppressed T2-weighted imaging can be useful for identifying osseous pathology in patients with painful scoliosis. Unless scoliosis is suspected to be caused by a spinal mass, contrast is not usually indicated (ACR Practice). Our contextualized MRI spine protocol for scoliosis includes a coronal T2 sequence with fat suppression for the detection of segmentation anomalies, bone lesions, and paraspinal masses, as well as an axial T1-weighted sequence for the lumbar spine to detect fatty terminal filum lesions, which could result in scoliosis when they are associated with cord tethering (Table 4) (Figure 6).

Chiari/CCJ. The majority of imaging indications coded as “Chiari” were likely Chiari I, and there were additional MRI indications for the assessment of the craniocervical junction (CCJ), which are mostly for Chiari I (Figure 1). Chiari I has a prevalence of 3.6% in children [24], while Chiari II only has an incidence of about 1 in 1000 live births [25].

The goal of the MRI assessment is typically to assess the presence of a Chiari I deformity to evaluate symptoms known to be associated with this condition, or to aid in surgical decision-making and planning if a Chiari I deformity is identified. Diagnostic imaging criteria for Chiari I deformity in children include tonsillar ectopia by 5 mm below the foramen magnum, peg-like deformity of the tonsils, obstruction of the surrounding CSF spaces, compression of the cord with inferior displacement of the obex, presence of syringomyelia, and retroversion of the odontoid process [26]. Surgical decisions may be affected by the presence of CSF flow restriction and abnormal motion of the intraspinal contents on phase contrast (PC) MR imaging [27].

Our contextualized MRI cervical spine protocol for the CCJ includes sagittal T1-weighted images to assess for the presence of diagnostic anatomical features of Chiari I deformity, as well as a 3D heavily T2-weighted sagittal sequence to evaluate restriction of the CSF spaces at the CCJ. We also include a PC CSF flow study, which allows additional evaluation of CSF flow dynamics and abnormal motion of intraspinal contents (Figure 7).

Trauma. The incidence of cervical spine injuries in the pediatric trauma population is 1.0–1.3% [28]. Pediatric patients have different injury patterns than adults, especially under the age of 8 years, due to underossification and lax soft tissues related to immaturity. As a result, soft tissue injuries are more common in young children and can be missed on radiographs and CT imaging [29]. Detection of soft tissue injuries is important as up to 40% require surgical spine interventions, such as halo placement or internal fixation [30]. MRI for spinal trauma is usually appropriate in patients who are obtunded or nonverbal, in symptomatic patients with equivocal radiographs or CT imaging, and inpatients with neurologic findings in the absence of radiological findings. MRI has been suggested for those children in whom unconsciousness is predicted to last beyond 48 h or in whom clinical clearance within 72 h is unlikely [29].

MRI findings that can be seen with cervical spine injury include cord injury, intraspinal hemorrhage with or without mass effect, bone marrow edema and vertebral height loss, abnormal bony alignment, and soft tissue edema. Typical imaging sequences include sagittal and axial T2-weighted images with fat suppression, sagittal T1-weighted images to evaluate anatomy and subacute blood products [31].

Our contextualized MRI cervical trauma protocol includes sagittal T1-weighted images to assess for alignment and the presence of extramedullary blood products, an axial gradient echo sequence for visualization of intramedullary blood products, as well as a 3D heavily T2-weighted sagittal sequence to evaluate for ligamentous injuries (Table 5) (Figure 8).

Non-accidental trauma. There is scientific evidence of spine injury in children with non-accidental trauma. Pathologies seen on MR imaging include ligamentous injury at the craniocervical junction in up to 78% and spinal subdural hemorrhages in up to 68% of patients being evaluated for non-accidental trauma [32]. It has therefore been advocated to include total spinal MRI in the work-up for non-accidental trauma patients [33,34].

Our contextualized MRI non-accidental trauma protocol includes sagittal T2-weighted images with fat suppression to assess for the presence of ligamentous injury, intraspinal pathology, and any bone edema (Figure 9).

## 4. Discussion

It is not effective for radiologists to attempt to memorize the multitude of contextualized MRI protocols that we employ at our institution. Our protocol process is driven by our electronic health record system (EPIC Hyperspace^®^, Radiant, version May 2022, EPIC System, Software Company, Verona, WI, USA.) (Figure 10). Based on the clinical indication and requested procedure code, the radiologist can easily select from a menu of contextualized spine protocols. The radiologist is able to further customize these protocols as needed with specific instructions around the area of coverage, use of IV contrast, or added sequences. Additional specific narrative comments to the technologists can be added as well. It is important to note that successful imaging relies on patients remaining still. Sedation or general anesthesia is commonly administered for children at our institution who are determined to most likely not be able to hold still for the duration of the scan. In many cases, an attempt without sedation or anesthesia may be performed, and the scan is rescheduled with sedation or anesthesia if initially unsuccessful. Another factor to consider is the time required for contextualized MRI protocols. The time required for each protocol is variable, primarily dependent on the scanner being used and patient size. Our experience is that the protocols we employ are obtainable in a clinically feasible amount of time. Since the implementation of these protocols at our institution, we have not noticed any significant change in degree of motion artifact, nor have we experienced pushback from colleagues in anesthesia or sedation. Some of our protocols are designed to reduce total scan time, for example, in the case of the drop metastases protocol, which omits sequences such as pre-contrast T1 and axial T2 imaging, which we felt do not add value to the assessment for leptomeningeal metastases. This reduction in scan time is particularly helpful, given how long the MRI brain and spine procedure can be for patients presenting for brain tumor follow-up.

## 5. Conclusions

In summary, pediatric spine imaging can significantly benefit from MRI protocols tailored to specific pediatric indications. The contextualized pediatric spine MRI protocols presented in this paper can be adopted or adapted to institutional needs to better serve pediatric populations.

## Figures and Tables

**Figure 1 diagnostics-13-02973-f001:**
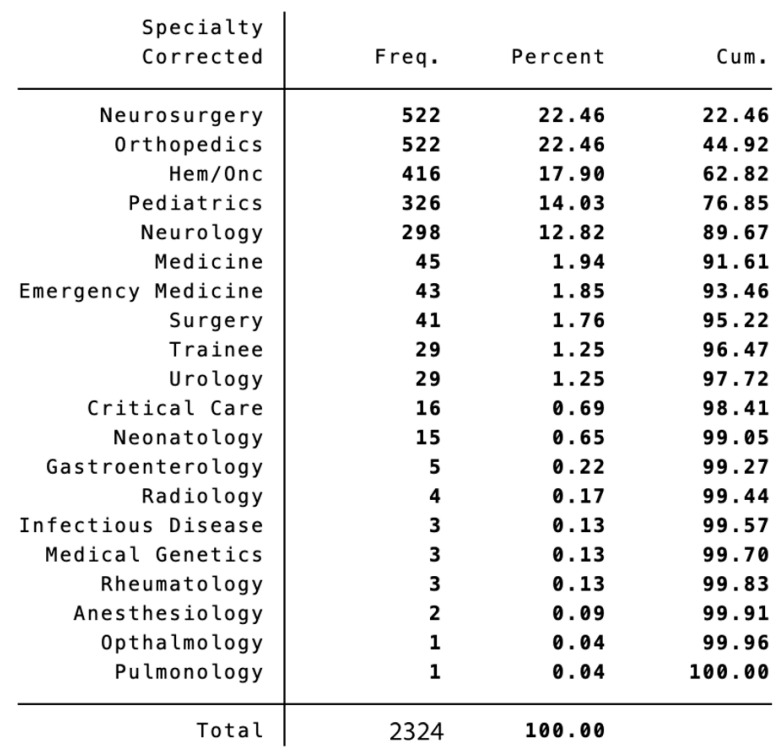
Ordering providers. Among the top ten referring provider specialties for pediatric spine imaging were Neurosurgery, Orthopedics, Hematology/Oncology, Pediatrics, and Neurology. Hem/Onc = Hematology/Oncology; Freq. = Instances in the data set; Cum. = Cumulative percentage.

**Figure 2 diagnostics-13-02973-f002:**
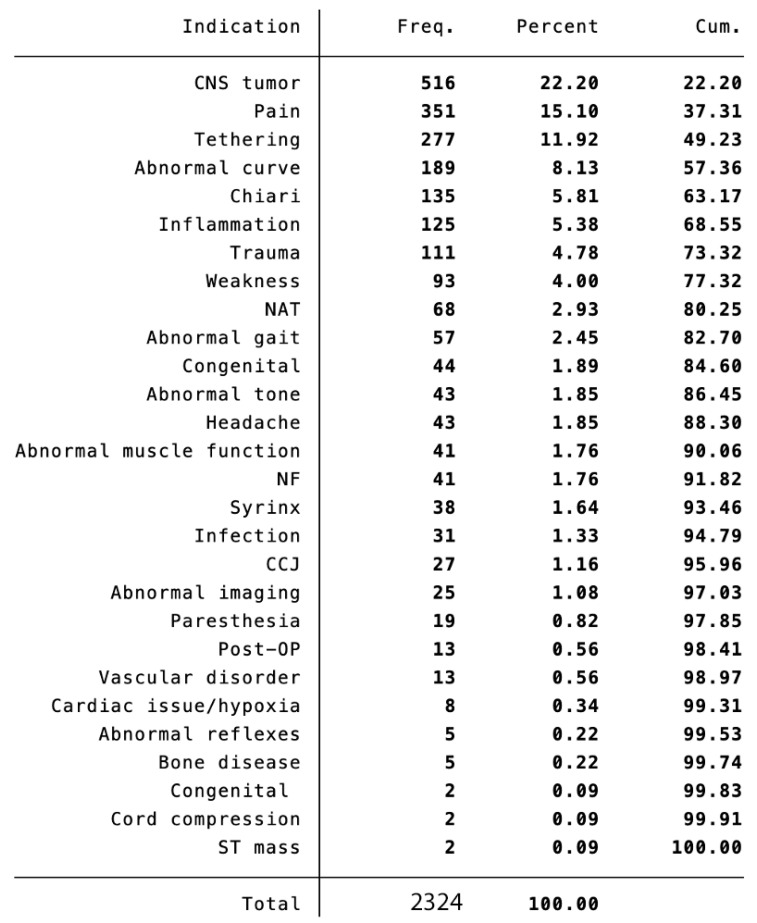
Imaging Indications. Among the top ten indications were CNS tumor, pain, cord tethering, abnormal spinal curve, Chiari I or II, inflammation, trauma, weakness, non-accidental trauma, and abnormal gait. CNS = Central nervous system; NAT = Non-accidental trauma; NF = Neurofibromatosis; CCJ = Craniocervical junction; Post-OP = Post-operative; ST = Soft tissue; Freq. = Instances in the data set; Cum. = Cumulative percentage Contextualized Imaging Protocols.

**Figure 3 diagnostics-13-02973-f003:**
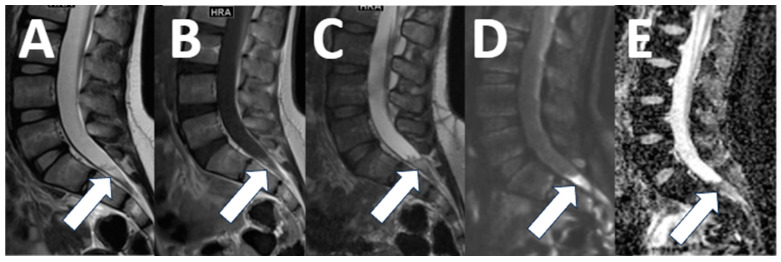
Spinal metastatic disease. An 8-year-old male with a newly diagnosed thoracic intramedullary tumor that was determined to be an ependymoma on subsequent surgical pathology. The conventional sagittal T1-weighted images show some inhomogeneous signals in the thecal sac ((**A**) arrow), with subtle contrast enhancement on sagittal T1-weighted imaging ((**B**) arrow). The sagittal 3D T2 acquisition shows definite soft tissue ((**C**) arrow) with increased diffusion ((**D**), arrow) and diffusion restriction ((**E**), arrow), consistent with metastatic disease.

**Figure 4 diagnostics-13-02973-f004:**
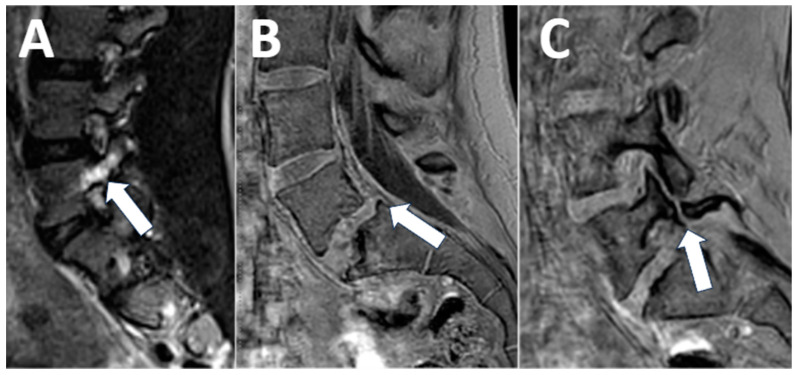
Spondylolysis. (**A**) A 12-year-old female with back pain. The sagittal T2-weighted image with fat saturation shows left-sided pedicle edema ((**A**), arrow). (**B**,**C**) A 14-year-old female, competitive swimmer with back pain. Sagittal 3D T1 imaging shows anterolisthesis of L5 on S1 ((**B**), arrow) as well as a corticated fracture of the L5 pars articularis consistent with spondylolysis ((**C**), arrow).

**Figure 5 diagnostics-13-02973-f005:**
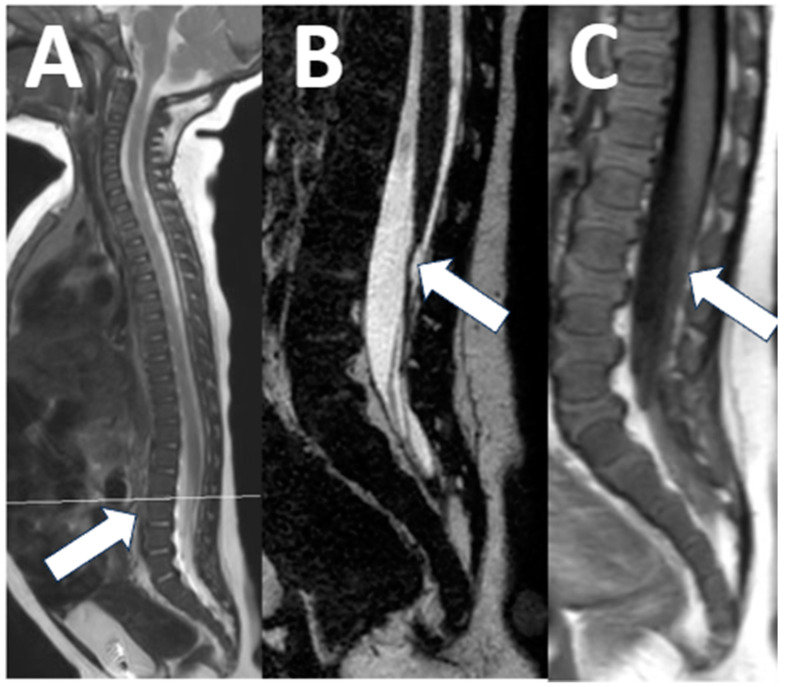
Cord tethering. A 4-month-old male with an abnormal gluteal cleft and subsequently a reported low-lying conus on ultrasound. The MRI shows a normal location of the conus at the L2–3 disc level when cross-referencing the axial images with the sagittal view and counting the vertebral bodies from top to bottom ((**A**), arrow). There was, however thickening of the terminal filum on sagittal 3D T2 images ((**B**), arrow) without a corresponding T1 fat signal ((**C**), arrow), consistent with a thickened terminal filum. The patient was clinically diagnosed with cord tethering and surgical options were discussed.

**Figure 6 diagnostics-13-02973-f006:**
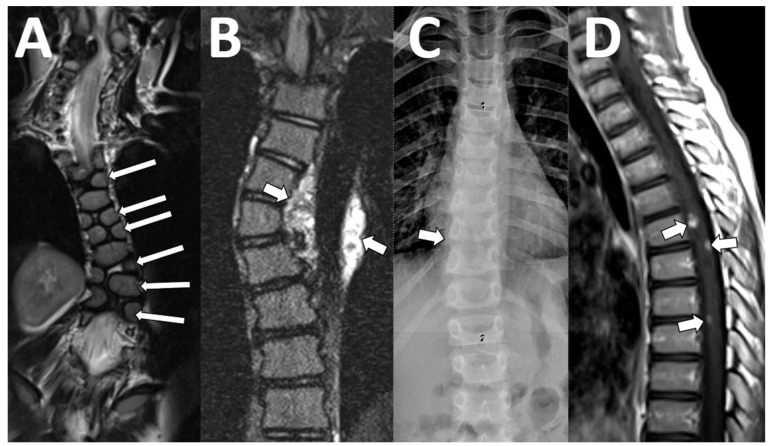
Scoliosis. (**A**) A 21-month-old male with infantile scoliosis. The coronal T2 images show a levoconvex thoracic curve and multilevel vertebral body segmentation abnormalities ((**A**), arrows). (**B**) A 13-year-old female with neurofibromatosis type 1 and scoliosis. There is a dextroconvex thoracic curve caused by a left paraspinal plexiform neurofibroma that also partially encases the descending aorta ((**B**), arrows). (**C**,**D**) A 9-year-old male with mild new scoliosis on radiographs ((**C**), arrow). Post-contrast T1-weighted images demonstrate an intramedullary mass without several foci of contrast enhancement ((**D**), arrows), which was subsequently diagnosed as a ganglioglioma on surgical pathology.

**Figure 7 diagnostics-13-02973-f007:**
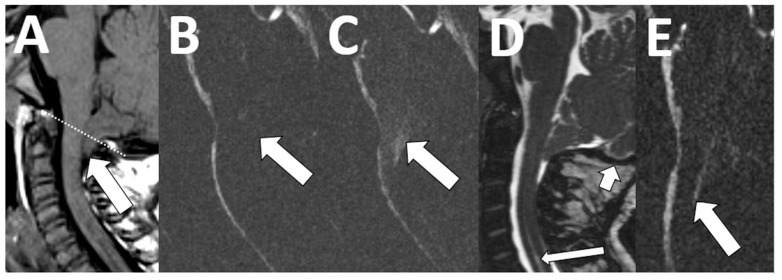
Chiari deformity. An 18-month-old female with new onset seizures and speech delay. Sagittal T1-weighted image shows descent of the cerebellar tonsils ((**A**), arrow) below the foramen magnum ((**A**), dotted line) by 8 mm. There was diminished flow across the posterior craniocervical junction ((**B**), arrow) and abnormal motion of the tonsils ((**C**), arrow). Follow-up imaging at 4 years of age shows interval suboccipital decompression surgery ((**D**), short arrow) and improved CSF flow across the craniocervical junction ((**E**), arrow), as well as resolved abnormal tonsillar motion. A small cervical syrinx persists post-surgically ((**D**), long arrow).

**Figure 8 diagnostics-13-02973-f008:**
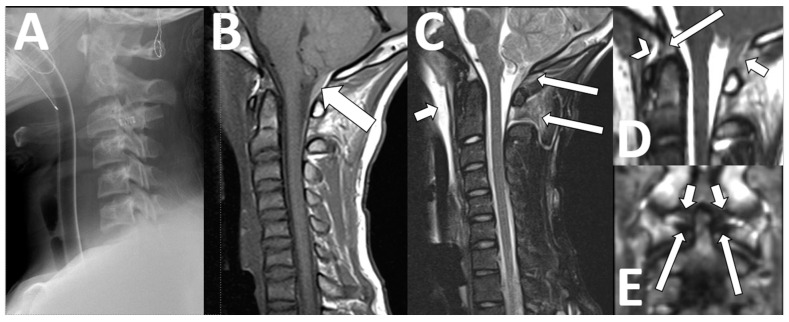
Cervical trauma. A 14-year-old male with head injury after MVC. (**A**) An initial cervical radiograph to the C6 level showed no spinal fracture or subluxation. (**B**) Sagittal T1-weighted imaging best shows an epidural hemorrhage at the posterior craniocervical junction that was known from prior head CTs ((**B**), arrow). The fat-suppressed sagittal T2 best shows soft tissue injuries, such as prevertebral swelling ((**C**), short arrow) and edema in the inter- and supraspinous soft tissues from the occiput to the C2–3 disc level ((**C**), long arrows). The 3D T2-weighted images better show rupture of the posterior atlanto-occipital membrane ((**D**), short arrow), and non-visualization of both the anterior atlanto-occipital membrane ((**D**), arrowhead) and the apical ligament ((**D**), long arrow). Coronal reformats of the 3D T2-weighted images show intact alar ((**E**), short arrows) and transverse ((**E**), long arrows) ligaments.

**Figure 9 diagnostics-13-02973-f009:**
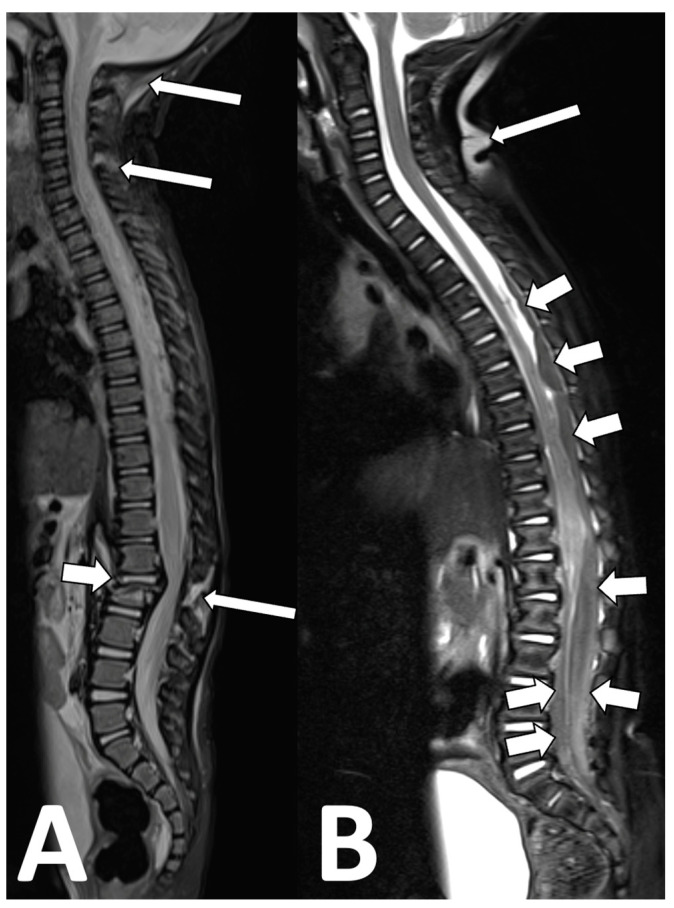
Non-accidental trauma spine imaging. (**A**) A 23-month-old male with suspected non-accidental trauma. The fat-suppressed sagittal T2 images show an L2 vertebral fracture with mass effect on the conus ((**A**), short arrow), as well as multiple levels of interspinous ligament injury at C1–6 and at L2–3 ((**A**), long arrows). Of note, using an inversion recover (STIR) technique for fat suppression in this case limits the evaluation of the spinal cord and spinal canal. (**B**) An 8-month-old male with suspected non-accidental trauma. Here, a fat-saturation technique was used, which improves the visibility of the spinal cord and identification of large intradural hematomas compressing the cauda equina ((**B**), short arrows). A downside of fat saturation is the inhomogeneity of the fat suppression, with areas of failed fat suppression ((**B**), long arrow).

**Figure 10 diagnostics-13-02973-f010:**
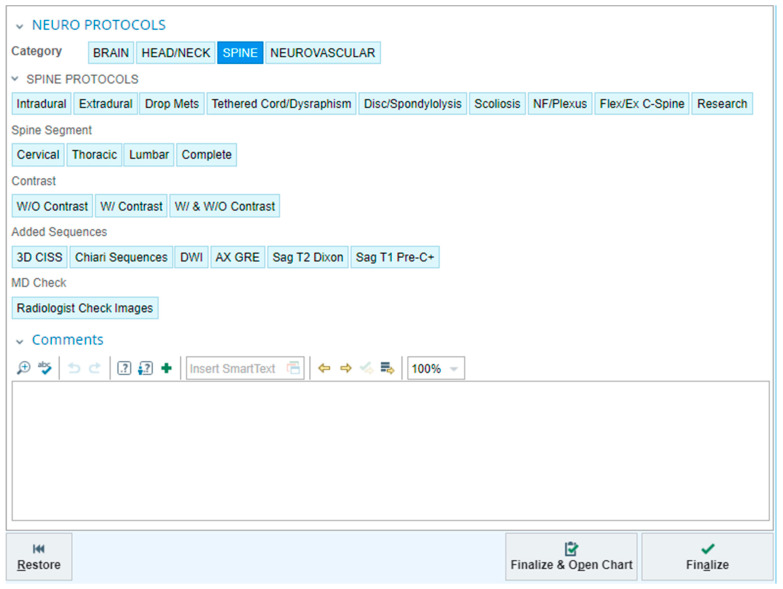
Protocol function in our electronic health record system. After selecting the “SPINE” category (top row), the radiologist can choose from a variety of contextualized protocols, select the anatomy, then specify the use of IV contrast. Additional sequences and free text comments can be added as needed.

**Table 1 diagnostics-13-02973-t001:** Spinal metastatic lesions.

Sequence	Plane	Slice Thickness/Gap (mm)	Tech Notes
3D CISS C+	SAG	0.60–0.75 mm gap 0%	Isometric voxels. Complete spine
	AX reformat	2.0/0.0	
T1 FLAIR C+ on 3T T1 TSE C+ on 1.5T	SAG	3.0/0.3	Complete spine
3D T1 VIBE C+	AX	3.0–4.0/0.0	Complete spine
Optional sequences			
DWI RESOLVE/ZOOM C+	SAG	3.0/0.3	Radiologist to add in cases of DWI restricting primary brain tumors or can use as a problem-solving tool.
3D T2 FLAIR C+	SAG	1.0/0.0 Isotropic voxels	Complete spine

**Table 2 diagnostics-13-02973-t002:** Disc/Spondylolysis Protocol for pain.

Sequence	Plane	Slice Thickness/Gap (mm)	Tech Notes
ENTIRE SPINE VERTEBRAL COUNT	SAG	4.0–5.0/0.8–1.0	Required only if the C-spine has NOT been ordered as part of the study
3D T1 VIBE/SPGR	SAG	1.0/0.0	Just through vertebral bodies of interest
T2 TSE	SAG	3.0/0.3	
T2 STIR	SAG	3.0/0.3	
T2 TSE	ANGLED AX	3.0/0.0	Angled through each disc in area of interest
Optional contrast			
T1 DIXON C+	SAG	3.0/0.3 mm gap 10%	
T1 DIXON C+	AX	4.0–5.0/0.4–0.5	Through entire spine section being scanned. Not angled to the discs

**Table 3 diagnostics-13-02973-t003:** Tethered Cord/Dysraphism.

Sequence	Plane	Slice Thickness/Gap (mm)	Tech Notes
ENTIRE SPINE VERTEBRAL COUNT	SAG	4.0–5.0/0.8–1.0	Required only if the C-spine has NOT been ordered as part of the study
T2 HASTE	COR C-T and/or T-L	5.0/0.5	Do only in area of spine ordered
T2 TSE	SAG	3.0/0.3	
T1 FLAIR (3T) T1 TSE (1.5T)	SAG	3.0/0.3	
T1 3D VIBE	AX	3.0–4.0/0.0	From end of cord to tip of coccyx AND through any dysraphic or fatty mass at any other scanned spine level; Do not do through entire spine
3D SPACE or CISS	SAG	0.60–0.75/0.0	Cover entire spine if ordered as complete spine or TL spine
	AX Reformat	Isotropic	
	COR Reformat	Isotropic	
Optional contrast			
T1 TSE FS C+	SAG	3.0–4.0/0.3–0.4	
T1 TSE FS C+	AX	4.0–5.0/0.4–0.5	
Optional sequence			
DWI RESOLVE	SAG	3.0–4.0/0.3–0.4	If concern for dermoid/epidermoid; Cover only the region of interest

**Table 4 diagnostics-13-02973-t004:** Scoliosis (* indicates notes regarding anatomical coverage).

Sequence	Plane	Slice Thickness/Gap (mm)	Tech Notes
T2 HASTE	COR	4.0–5.0/0.8–1.0	CT and TL
T1 FLAIR on 3T T1 TSE on 1.5T	SAG	3.0–4.0/0.3–0.4	Total spine * Scan to tip of coccyx
T2 TSE	SAG	3.0/0.3	
T2 TSE	AX	4.0–5.0/0.4–0.5	
T1 VIBE	AX	3.0/0.3	L spine only * From end of cord to tip of coccyx

**Table 5 diagnostics-13-02973-t005:** Cervical Spine Trauma. Our radiologist used the extradural spine protocol and added sagittal CISS and axial gradient echo imaging. Of note, the neuroradiologist can add additional sequences, such as axial GRE in the EMR protocol menu.

Sequence	Plane	Slice Thickness /Gap(mm)	Tech Notes
ENTIRE SPINE VERTEBRAL COUNT	SAG	4.0–5.0/0.8–1.0	Required if the C-spine has NOT been ordered as part of the study
T1 FLAIR (3T)/ T1 TSE (1.5T)	SAG	3.0/0.3	
T2 DIXON	SAG	3.0/0.3	
T2 DIXON	AX	4.0–5.0/0.4–0.5	Slice thickness depending on size of patient; cover vertebrae and paraspinous soft tissues
Optional sequence			
DWI RESOLVE	SAG	5.0/0.5	MD to order if indicated, limit to area of interest (i.e., not whole spine)
Optional contrast			
T1 DIXON	SAG	3.0/0.3	
T1 DIXON	AX	4.0–5.0/0.4–0.5	Slice thickness depending on size of patient; cover vertebrae and paraspinous soft tissues

## Data Availability

The data presented in this study are available on request from the corresponding author. The data are not publicly available due to HIPAA/privacy.

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
