# Peer review of "Diagnostic Excellence in Pediatric Spine Imaging: Using Contextualized Imaging Protocols"

_diagnostics, 2023, doi:10.3390/diagnostics13182973_

Round 1
Reviewer 1 Report
Dear Author, the idea behind the study is very good. one thing if you could highlight, it will be better. what is your plan for those pediatric spine patients who are not cooperative enough for MRI and are ultimately need some sort of sedation or general anaesthesia ?
Author Response
Thank you- we addressed the comments

Reviewer 2 Report
Dear Authors,
You presented a very thorough study on spine imaging. The study is important as the pediatric population poses particular challenges due to both the age-related pathology as well as the difficulty of performing a quality MRI exam at their age. The study goes through almost all aspects of pediatric spine pathology from indications to protocol contextualizing to best fit the information needed in every case, with appropriate and clear imaging.
However, there are a couple of important issues that need to be resolved before I can recommend it for publication.
1. Title: having CNS in there is misleading, as the whole study deals with the spine and its pathology. It needs to be reformulated in order to mirror the content.
2. Figures 1 and 2 are introduced in the opposite order in the text. Also, starting the main body of the paper with a figure with no introduction is not common and somehow misleading.
3. There is little mention of the duration of the studies and how the contextualized approach changes the time in the MRI. This is an important issue for the age group we are talking about and how the authors addressed it is of interest to every reader.
4. Two sections are missing completely: Discussion and Conclusions. They have to be here, especially since you are proposing a change of paradigm and point of view. Also, the discussion would be a good place to compare time requirements and eventual measures to address them if they arrise.
The English language is mostly correct with minor typos.
Author Response
Thank you- we addressed the comments

Round 2
Reviewer 2 Report
Dear Authors,
The study looks and feels more consistent and cohesive from title to conclusion. I would change the name of the section "Operational considerations" to "DIscussion" but either than that, I recommend the publication in the current form.